# Gateway-Free LoRa Mesh on ESP32: Design, Self-Healing Mechanisms, and Empirical Performance

**DOI:** 10.3390/s25196036

**Published:** 2025-10-01

**Authors:** Danilo Arregui Almeida, Juan Chafla Altamirano, Milton Román Cañizares, Pablo Palacios Játiva, Javier Guaña-Moya, Iván Sánchez

**Affiliations:** 1Facultad de Hábitat, Infraestructura y Creatividad, Pontificia Universidad Católica del Ecuador, Quito 17012184, Ecuador; dsarregui@puce.edu.ec (D.A.A.); jchafla390@puce.edu.ec (J.C.A.); mnroman@puce.edu.ec (M.R.C.); eguana953@puce.edu.ec (J.G.-M.); 2Escuela de Informática y Telecomunicaciones, Universidad Diego Portales, Santiago 8370190, Chile; pablo.palacios@mail.udp.cl; 3Department of Networking and Telecommunication Engineering, Universidad de las Américas, Quito 170503, Ecuador

**Keywords:** ESP32-S3, SX1262, IoT, LoRa mesh, listen-before-talk, self-healing, hop-by-hop ACK

## Abstract

LoRa is a long-range, low-power wireless communication technology widely used in Internet of Things (IoT) applications. However, its conventional implementation through Long Range Wide Area Network (LoRaWAN) presents operational constraints due to its centralized topology and reliance on gateways. To overcome these limitations, this work designs and validates a gateway-free mesh communication system that operates directly on commercially available commodity microcontrollers, implementing lightweight self-healing mechanisms suitable for resource-constrained devices. The system, based on ESP32 microcontrollers and LoRa modulation, adopts a mesh topology with custom mechanisms including neighbor-based routing, hop-by-hop acknowledgments (ACKs), and controlled retransmissions. Reliability is achieved through hop-by-hop acknowledgments, listen-before-talk (LBT) channel access, and duplicate suppression using alternate link triggering (ALT). A modular prototype was developed and tested under three scenarios such as ideal conditions, intermediate node failure, and extended urban deployment. Results showed robust performance, achieving a Packet Delivery Ratio (PDR), the percentage of successfully delivered DATA packets over those sent, of up to 95% in controlled environments and 75% under urban conditions. In the failure scenario, an average Packet Recovery Ratio (PRR), the proportion of lost packets successfully recovered through retransmissions, of 88.33% was achieved, validating the system’s self-healing capabilities. Each scenario was executed in five independent runs, with values calculated for both traffic directions and averaged. These findings confirm that a compact and fault-tolerant LoRa mesh network, operating without gateways, can be effectively implemented on commodity ESP32-S3 + SX1262 hardware.

## 1. Introduction

The Internet of Things (IoT) is rapidly expanding in the consumer and industrial domains, aiming to automate and optimize processes through sensors and actuators typically managed by microcontrollers [1,2,3]. Common communication options such as Wi-Fi, Bluetooth, Zigbee, or wired links are often limited in range and require complex infrastructure [4,5,6,7]. Consequently, the industry is turning to long-range wireless solutions, such as Long Range (LoRa), which offer reliable connectivity in large-scale, rural, or infrastructure-constrained environments [8].

LoRa stands out as one of the most popular wireless technologies in IoT applications due to its low energy consumption, transmission range, and strong ability to penetrate obstacles [9,10]. It operates in unlicensed sub-GHz frequency bands, typically 433 MHz, 868 MHz, and 915 MHz, depending on regional regulations [11]. LoRa uses the Chirp Spread Spectrum (CSS) as its modulation technique, in which the information is encoded using chirps, or frequency-modulated signals that increase or decrease frequency over a defined time interval [12]. This modulation technique provides high resistance to interference and noise, making LoRa particularly suitable for low-power long-range communications in challenging environments [13].

Furthermore, LoRa operates exclusively at the physical layer of the Open Systems Interconnection (OSI) model, which means that it does not incorporate mechanisms for Medium Access Control (MAC), device addressing, or session management [14]. As a result, packet collisions are more likely to occur, especially in dense networks [15,16]. To overcome these limitations and fully exploit LoRa’s potential, it is often combined with LoRaWAN, a higher-layer protocol that builds on LoRa as its physical layer foundation while providing essential network functionalities such as device management, routing, medium access control, and other communication services [12].

Despite the advantages provided by LoRaWAN, its adoption presents several limitations, particularly when targeting low-cost microcontroller-based IoT deployments. The protocol is inherently designed around a star-of-stars topology, where communication between the end devices and the network servers is entirely dependent on the gateways [17,18]. This architecture introduces higher infrastructure costs and creates a potential single point of failure, as each gateway becomes a critical dependency on the communication chain [19].

Moreover, although LoRaWAN is a widely adopted protocol compatible with commercial microcontrollers such as ESP32, ESP8266, and ATmega328, it relies on centralized gateways for communication between end devices and network servers [20,21,22,23]. This dependency introduces financial and architectural complexity, particularly in scenarios where the deployment of a full infrastructure is impractical. Although these microcontrollers are ideal for low-cost IoT networks due to their efficiency and affordability, their use in LoRaWAN remains constrained by this centralized architecture [24,25].

Several studies have investigated multi-hop LoRa networks to overcome the limitations of traditional star-topology LoRaWAN deployments. These works often focus on aspects such as improving energy efficiency, enhancing scalability, optimizing coverage, developing medium access control strategies, or targeting specific applications and use cases, through the use of simulation models or optimized routing protocols [26]. Although these contributions demonstrate the feasibility of mesh-like architectures over LoRa, they are generally designed for abstract or specialized hardware environments. In contrast, little emphasis has been placed on implementing practical decentralized communication systems that operate directly on low-cost, commercially available microcontrollers accessible to general users, such as the ESP32 family.

Among the related works, the proposal presented by Bor et al. [27] stands out as one of the earliest to explore a multi-hop LoRa network through the introduction of LoRaBlink, a system focused on energy-efficient data collection across wide areas. This approach used a slotted transmission pattern to reduce collisions and power consumption, combined with flooding techniques for routing. Another alternative is Meshtastic, a popular open-source project designed for mesh-based communication between heterogeneous nodes, oriented primarily toward offline peer-to-peer messaging [28]. Furthermore, ref. [29] discusses a gateway-centric mesh routing topology, evaluated through an OMNeT++ simulation model, which focuses on packet delivery analysis in a network where multiple mesh-connected nodes communicate through a single gateway.

A comparison between these works and the present system is provided in Table 1, highlighting differences in target hardware, routing approach, reliability mechanisms, channel access, code availability, evaluation type, and energy reporting.

Therefore, developing alternative networking protocols or lightweight communication systems that enable direct, decentralized interaction among microcontrollers without requiring fixed infrastructure would be highly advantageous. Such approaches could reduce deployment costs, improve scalability, and improve the resilience of IoT networks in constrained or remote environments.

In addition, the system implements a custom networking stack that spans the Physical (PHY), Data Link, and Network layers, while deliberately leaving the Application layer to users. By design, the architecture is gateway-free, enabling direct peer-to-peer and multi-hop communication among nodes without any centralized infrastructure. This design keeps the system lightweight and practical for ESP32-based hardware.

To address these limitations, this work implements a mesh-networking approach built directly on top of the physical layer of LoRa, eliminating the need for centralized gateways. In such a topology, all nodes are capable of forwarding messages on behalf of others, enabling multi-hop communication and increasing the reliability of the network against node or link failures [30]. This decentralized design ensures that, even in the presence of individual node failures or changes in the topology, other nodes can still communicate through alternative paths.

In this paper, a mesh-based communication system is presented, built directly on the LoRa physical layer, and specifically designed for microcontroller platforms such as the ESP32, which are widely used in IoT applications due to their affordability, accessibility, and integrated wireless capabilities. The system enables decentralized, multi-hop communication between nodes, aimed at enhancing network resilience and minimizing infrastructure costs in deployments based on commonly available microcontroller hardware. Key contributions include the gateway-free mesh implementation on ESP32, a lightweight protocol with neighbor-based routing and hop-by-hop acknowledgments, self-healing mechanisms for automatic failure recovery, and empirical validation across multiple test scenarios.

To assess its basic functionality and validate the viability of the design, a series of initial tests were performed. These evaluations focused on the packet delivery ratio, fault recovery capacity, and the number of retransmissions required under varying network conditions.

The primary aim is to validate the minimal stack, communication capabilities, and self-healing behavior under controlled test conditions. Accordingly, the present results must not be interpreted as production readiness in untrusted settings, as this work presents a preliminary implementation intended to validate and test the proposed design concept, without incorporating the system optimization, energy efficiency measures, and security features necessary for production environments, which were out of scope.

To better position this contribution within the existing literature, key technologies and design aspects addressed by previous work are highlighted, namely (i) multi-hop strategies based on flooding such as LoRaBlink [27], which prioritize energy efficient data collection through slotted transmissions; (ii) mesh-oriented frameworks such as Meshtastic [28], which enable peer-to-peer communication but remain focused mainly on messaging applications; and (iii) gateway-centric approaches [29], which optimize routing and delivery but retain centralized dependencies. These works collectively demonstrate the feasibility of extending LoRa networks beyond the star topology; however, they do not provide a lightweight, fully decentralized solution directly implementable on widely accessible microcontrollers, such as the ESP32.

The remainder of this paper is organized as follows. Section 2 details the lightweight LoRa mesh stack and loop prevention logic supported by a concise state machine and describes the firmware architecture and implementation on ESP32-S3 + SX1262. This section also presents the hardware platform, antenna considerations, and a link budget summary and finally outlines the experimental methodology. Section 4 reports the results in ideal, failure, and urban settings and then discusses design trade-offs, co-existence, and parameter sensitivity. Section 5 details the limitations and threats to the validity of this study. Finally, Section 6 presents the conclusions and describes future work. The Appendix A provides additional tables and supplementary equations.

## 2. Materials and Methods

The implemented communication system consists of a decentralized architecture based on LoRa mesh networking, specifically designed for low-cost and commercially available microcontroller platforms. In this scheme, all nodes on the network share the same role and functionality. They can operate as transmitters, receivers, and repeaters, enabling multi-hop packet propagation without relying on centralized infrastructure or specialized hardware. Communication is established through single-hop links between neighboring nodes, while end-to-end connectivity is achieved via a neighbor-based routing strategy in which packets are forwarded to the most suitable adjacent node.

The system operates across the physical, data link, and network layers of the OSI model, excluding upper-layer functionalities such as application-level protocols, which are left to the user’s implementation or adaptation. The architecture is supported by a custom firmware stack developed in C++ for the ESP32 family of microcontrollers, managing packet structure, routing logic, transmission scheduling, and reception handling.

### 2.1. Hardware Platform

The hardware platform used in this work is the Heltec Wireless Stick V3 (ESP32-S3FN8 + SX1262, Heltec Automation, Chengdu, China), a compact and versatile development board designed for low-power, long-range wireless communication. It operates in the 433–928 MHz sub-GHz bands, making it suitable for various LoRa applications under different regional regulations.

As shown in Figure 1, the board exposes a wide range of digital and analog pins, enabling easy integration of external sensors, actuators, and power sources. These include interfaces for SPI, UART, ADC, DAC, and battery connection, which makes it suitable for embedded IoT applications that require both connectivity and peripheral interaction.

Table 2 provides a summary of the main technical specifications of the board, including details of the ESP32-S3 microcontroller, the SX1262 transceiver, the OLED display onboard, memory resources, and power management features.

Regarding the external antenna, a Heltec-supplied glue rod monopole was used (Glue Rod, Heltec Automation, Chengdu, China). The dedicated IPEX (U.FL) antenna interface of the board for LoRa was connected to the glue rod antenna through an IPEX (U.FL) Ver.1 to the SMA feeder cable. The main antenna characteristics are summarized in Table 3.

In addition, a prototype case was designed using Tinkercad (Autodesk Inc., San Francisco, CA, USA) and printed in Polylactic Acid (PLA) with an ANET 3D printer (ET4 Pro, Shenzhen Xiangli Industry Co., Longhua District, Shenzhen, China). The enclosure aims to protect the development board during handling and operation, preventing internal damage. Its design prioritizes practicality and functionality over aesthetics.

### 2.2. System Design

#### 2.2.1. Packets Definitions

The system defines several types of packets to support the core communication mechanisms, such as message forwarding, acknowledgment, neighbor discovery, and route adaptation. All packets share a common header that enables consistent processing between nodes. The summary of the structure of the packet is available in Table A1, Appendix A.

Each type includes additional fields tailored to its role. DATA packets carry fields for data delivery. ACK packets confirm reception, HELLO packets announce node presence, and ALT packets suggest rerouting after detecting duplicates. Table A2, Appendix A summarizes their structure and function.

Taking into account that the common header occupies 7 B, the serialized payload lengths are as follows: DATA 19 B, HELLO 9 B, and ACK/ALT 11 B (packed, no padding).

#### 2.2.2. Core Mechanisms

To enable reliable multi-hop communication over LoRa, the firmware integrates a set of distributed mechanisms responsible for routing, channel access, reliability, and autonomous behavior. These mechanisms operate in a decentralized manner and are designed to be functional and simple. A general overview is provided in Table A3, Appendix A.

For routing, the system relies on a neighbor-based strategy in which each node broadcasts a HELLO beacon at regular intervals of 60 s to announce its presence. Active neighbors are stored in a local table that contains the node ID, the Received Signal Strength Indicator (RSSI), and a timestamp of the last contact. Neighbor entries are automatically removed if no beacon is received within a defined expiration timeout of 120 s. The routing algorithm for next-hop selection depends on the calculated score for each neighbor, using a simple heuristic that favors recent and strong links.    (1)$$score=RSSI-\frac{t\text{\_}now-t\text{\_}lastHeard}{1000}$$
where *RSSI* is the last received signal strength (in dBm), *t_now* is the current system time in milliseconds, and *t_lastHeard* is the time when the last beacon was received from that neighbor.

Next-hop selection proceeds in two steps. First, the candidates are the neighbors currently present (i.e., not expired). If the final destination is in that set, the packet is forwarded directly without applying the score. Otherwise, the score is computed for all candidates, and the three with the highest values form a short-list. One of them is then chosen at random at transmission time to distribute load and exploit parallel routes.

For medium access control, the firmware implements an LBT mechanism to reduce the probability of collisions. Before any transmission, the node opens a 500-ms listening window to check channel activity. If no incoming packet is detected during that window, meaning no hardware interruption occurs, the transmission proceeds. Otherwise, a back-off strategy is applied by introducing a random delay between 500 and 1000 ms. This process may be repeated up to five times. This limits retries to five ensures that the node does not remain indefinitely blocked in high contention scenarios. If the limit is reached, the packet is sent even if the channel is busy, guaranteeing forward progress.

This policy is intentionally simple as it reduces collision likelihood without relying on additional channel probing. This approach was chosen because using RSSI thresholds to determine channel occupancy is inherently imprecise, as the RSSI value can vary significantly depending on the radio environment and the distance between nodes, making it an unreliable metric for clear channel assessment [33].

Reliability is enforced through a per-hop acknowledgment mechanism. After fowarding a DATA packet, the node waits for an ACK with a timeout of up to 15 s. If no response is received within this window, the transmission is re-tried up to three times. If all retries fail, the system assumes that the route is no longer valid. The failed neighbor is removed from the routing table, and the packet is re-enqueued to attempt delivery via an alternate next hop, triggering a self-healing mechanism that maintains connectivity even under node or link failures. The limit of the self-healing process directly depends on the valid neighbors. If no eligible neighbor remains the packet is discarded.

Additional mechanisms enhance autonomy and robustness. Each device generates its own address by folding the ESP32’s MAC address into a 16-bit identifier, enabling plug-and-play deployment. To uniquely identify each message and reduce the probability of collisions, the system uses a 32-bit messageID composed of three components: the message type (8 bits), the node identifier (16 bits), and a random 8-bit value generated at the moment of creation. This combination greatly reduces the likelihood of messageID collisions, even at higher traffic loads.

To prevent forwarding loops and reduce redundant traffic, nodes maintain a rolling buffer with the last 30 processed messageIDs. When a duplicate is detected, further forwarding is stopped and a brief advisory via an ALT packet is sent backward to suggest route recalculation upstream. This filtering logic requires around 120 bytes of memory and remains lightweight enough for commercial microcontrollers. In addition, a cumulative TTL limited up to 6 hops is included in each packet and decremented at every hop, bounding the end-to-end delay and preventing infinite retransmissions.

The general operation of the mechanisms implemented and their interactions is summarized in the state machine diagram shown in Figure 2. This representation provides a high-level view of the transitions and conditions that govern the behavior of the system.

The state machine summarizes three normal entry paths: the node idles while listening and periodically emitting HELLO beacons; a local action requests the generation of a DATA packet; or a packet is received over the air. Upon reception, handling is driven by type: HELLO refreshes the neighbor table; DATA is delivered locally when the node is the final destination, otherwise it is relayed after next-hop selection and a listen-before-talk check; ACK ends the outstanding retry cycle associated with the corresponding DATA; and ALT indicates that the attempted path has already been visited upstream, so that the next hop is excluded from the eligible set and a new route is computed for the same DATA packet.

It is important to mention that all relevant timing, size, and behavior parameters can be edited through the configuration file (config.h), allowing flexible adaptation to different scenarios or use cases.

Finally, it is important to note that this system currently does not incorporate security mechanisms, as the focus was deliberately placed on validating the core mesh functionality and self-healing capabilities. Neither encryption nor authentication is applied to the transmitted packets. Although this may be acceptable in controlled test environments, it is not intended for practical deployments in untrusted or sensitive contexts.

In future work, lightweight encryption algorithms such as ASCON, GIFT, SIMON, and CryptoCore [34] will be evaluated, and an encryption-with-authentication approach will be explored, similar to the study performed by Iqbal and Iqbal [35], which employs similar technology and hardware. These future approaches will aim to ensure that the security layer does not significantly increase the computational load on the microcontroller platform.

### 2.3. Implementation

#### 2.3.1. Firmware

The node firmware was developed entirely in C++ using the Arduino IDE (v2.3.2, Arduino Srl, Monza, Italy), together with vendor-specific tools such as the board manager Heltec ESP32 Series Dev-boards (v3.0.2, Heltec Automation, Chengdu, China) and the libraryHeltec ESP32 Dev-Boards (v2.1.4, Heltec Automation, Chengdu, China).

The firmware follows a modular design, where each file encapsulates a specific system function, as detailed in Table 4.

Regarding the impact of firmware on hardware resources, in typical traffic (periodic HELLO beacons and bidirectional DATA + ACK exchanges) the firmware was used on average 39,328.8 B (~39.3 KB) of RAM and 383,264 B (~383.3 KB) of flash, the average CPU load was 1.14%. These values demonstrate that the prototype leaves ample headroom in memory and allows us to compute additional features. The per-run measurements are reported in Table A4, Appendix A.

The complete source code is available in the project public repository: https://github.com/Danar2714/LoRaMeshProject/tree/main (accessed on 20 July 2025), under the GNU General Public License v2.0 (GPL-2.0). The version used and tested in this paper is tagged v1.0 (commit d0ccf2a96e4c0e61935bb4d3a4d81826f579bf98).

#### 2.3.2. Prototype

The system was deployed in a basic mesh network prototype consisting of five nodes, each loaded with custom firmware. As shown in Figure 3, each node is enclosed in the 3D-printed protective case described in Section 2.1.

Regarding antenna placement, the external glue-rod monopole was connected through the board’s IPEX (U.FL) antenna interface via an IPEX (U.FL) Ver.1 to SMA feeder cable and mounted outside the 3D-printed PLA enclosure in a vertical orientation.

Consistent with Picha et al. [36], which identified higher and more frequency-dependent dielectric losses in pigmented PLA than in colorless PLA (1–100 MHz), and despite that study covering frequencies lower than the 915 MHz band used here, the observed trend motivated the placement of the radiator outside the red pigmented PLA enclosure and keeping it vertical. This placement also facilitates prototype disassembly.

Each node is powered by a USB Type-C connection, which also provides a serial interface to the development environment. This interface allows for real-time monitoring of communication logs, including received packets, debugging information, and error messages. Additionally, both received and transmitted DATA packets are displayed on the onboard OLED screen to provide immediate visual feedback to the user.

Moreover, to initiate the transmission of a DATA packet, the user must manually enter the target node ID through the serial monitor.

Regarding the prototype configuration, a partial mesh topology was adopted to ensure multi-hop communication between peripheral nodes while preserving alternative paths (Figure 4). To maintain a stable logical topology during evaluation, the firmware applied a compile-time neighbor-visibility mask (whitelist) that constrains routing eligibility after discovery. Dynamic neighbor discovery and maintenance remained active (HELLO/expiry; metric refresh), and faulty neighbors were evicted on missed HELLO packets or repeated ACK failures. For the prototype, the proposed routing and neighbor-maintenance mechanisms were implemented and used throughout all test runs.

All nodes were configured with the same radio parameters to ensure uniform testing conditions. Table 5 summarizes the key configuration values used during the prototype deployment, including frequency, transmission power, and modulation parameters. These settings were applied consistently across all test scenarios and can be modified via the config.h file in the firmware.

Spreading Factor (SF) 7 was selected because it delivers the highest physical-layer data rate and thus shorter packet duration. Most evaluation scenarios involved short inter-node distances, as detailed in Section 2.4.2, so this choice matched the experimental setup. Al-Sammak et al. explicitly select SF7 in urban evaluations due to its higher data rate [37]. Although SF7 is less suitable for long-distance coverage than higher spreading factors, transmission speed was the decisive criterion in the configuration of this prototype.

It is important to highlight that the selected radio parameters comply with current regulations in Ecuador [38], where the implementation and testing of the system were conducted. However, regulations establish that for real-world deployments in the 915–928 MHz band, Frequency Hopping Spread Spectrum (FHSS) is required and must be implemented in any production deployment. With respect to the duty cycle, local regulations specify that enforcement depends on the practical application. A 1/30 duty cycle is only mandatory for periodic-use applications. Therefore, future deployments of the prototype in the region should comply with Ecuador’s duty cycle regulations, depending on the intended usage.

Furthermore, in the current configuration, the Time-on-Air (ToA) values per packet type are as follows: DATA packets (19 B) have a ToA of 51.456 ms; HELLO packets (9 B) result in 41.216 ms; and ALT or ACK packets (11 B) produce 41.216 ms. These values were calculated using the formulas in Appendix A.

In addition, to contextualize the radio configuration, a single-link budget at 915 MHz was prepared using the parameters in Table 5 and the antenna in Table 3. Table 6 reports transmit power, feeder losses (based on cable insertion-loss [39]), antenna gains, receiver sensitivity (per the SX1262 datasheet [40]), and the Maximum Allowed Path Loss (MAPL) resulting from the chosen implementation and fade margins.

### 2.4. Testing Methodology

#### 2.4.1. Communication Metrics

The evaluation metrics were designed to assess the transmission and reception capabilities of the system, its self-healing behavior, and the reliability of its communication mechanisms. The objective is to verify the stability of communication, the resilience of the network to physical node failures, and the viability of the core mechanisms.

Packet Delivery Ratio (PDR): Percentage ratio between the total number of packets successfully received and the total number of packets transmitted. This metric assesses the overall effectiveness and robustness of communication.(2)$$\mathrm{PDR}=\frac{\mathrm{Received}\;\mathrm{Packets}}{\mathrm{Sent}\;\mathrm{Packets}}\times 100.$$

Packet Recovery Ratio (PRR): Percentage ratio between the number of packets successfully recovered during a network failure and the number of expected packets in that period. This metric evaluates the self-healing capabilities of the system.(3)$$\mathrm{PRR}=\frac{\mathrm{Recovered}\;\mathrm{Packets}}{\mathrm{Expected}\;\mathrm{Packets}}\times 100.$$

Total Retransmissions: Total number of retransmission attempts made before receiving an ACK, including retries across all intermediate hops. This metric is used to assess the efficiency and reliability of the hop-by-hop ACK mechanism.(4)$$\mathrm{Total}\;\mathrm{Retries}=\sum_{i=1}^{n}R_{i}$$
where $R_{i}$ is the number of retries for the *i*-th hop, and *n* is the total number of hops in the communication path.

#### 2.4.2. Testing Scenarios

To evaluate the prototype, three scenarios were defined: ideal conditions, induced failure conditions, and urban deployment. All scenarios share the same logical topology seen in Figure 4, where bidirectional traffic is injected from both ends of the communication path. For statistical consistency, each scenario comprises five independent runs under identical conditions, and the reported metrics are computed from the aggregate results across these repetitions.

As stated in Section 2.3.2, to maintain a stable and repeatable topology in all test scenarios, the set of valid neighbors was explicitly constrained in the firmware. Consequently, even if the physical arrangement of devices did not always coincide across runs or locations, the effective routing graph remained constant across scenarios.

The first scenario simulates ideal deployment conditions, where all nodes are active and located within short distances. In this configuration, multi-hop communication is feasible, including the availability of alternative paths and bidirectional messaging between peripheral nodes. Figure 5 depicts the scenario, where the purple line represents the route from node E to node A, and the green line indicates the reverse route.

The second scenario follows the same logical topology and node placement as the first, maintaining short distances. However, node D is intentionally disconnected to simulate a sudden node failure. As shown in Figure 6, communication between nodes E and A remains functional in both directions, demonstrating bidirectional packet delivery despite failure.

The third scenario adopts a topology similar to the first scenario, but with increased distances between nodes. Despite the extended range, bidirectional communication remains consistent throughout the test.

The urban tests were conducted on 24 May 2025, between 15:00 and 16:00 local time (America/Guayaquil, UTC-5) in Quito, Ecuador. Weather was partly cloudy with an approximate temperature between 10 °C and 17 °C. These observations provide context only and were not used to normalize or adjust the communication metrics.

Figure 7 shows the actual physical deployment of the nodes. Map created in QGIS (v3.44.1 “Solothurn”, QGIS Association, Grüt, Switzerland).

## 3. Results

### 3.1. Testing Scenarios Results

Data from the three scenarios were compiled into tables and processed to obtain PDR, packet-level PRR, and total retransmissions, computed per repetition and averaged per scenario. Each per-packet table also includes a Notes column that clarifies outcomes: ACK Timeout denotes a delayed or lost acknowledgment, Drop Packet indicates a loss after neighbor eviction due to missed HELLOs or an exhausted ACK wait, and Self-Heal marks rerouting once all retries and ACK waits on the initial path were exhausted.

#### 3.1.1. Scenario 1: No Failure Bidirectional Communication

The communication path between nodes A and E involved several hops, with each node acting as both transmitter and receiver during simultaneous bidirectional communication. Two packets were sent in each direction for each repetition, totaling four packets per repetition and twenty packets overall over five repetitions. In this scenario, the admissible routes between A and E are A–B–C–E or A–B–D–E; nodes C and D are parallel candidates, and when multiple neighbors are equally valid the routing algorithm breaks ties at run time via randomized selection, which does not affect the interpretation of delivery or retransmissions.

In four out of five repetitions, all packets were successfully delivered. Only the third repetition recorded a single lost packet. Table 7 presents the raw per-packet results for all repetitions.

After processing the results of each repetition, the overall mean PDR was 95% and the average number of retransmissions was 8 per repetition. These results and calculations are presented in Table 8.

#### 3.1.2. Scenario 2: Node Failure Bidirectional Communication

The configuration from Scenario 1 was replicated, but an intermediate node (D) was intentionally disconnected during the test execution to emulate a physical failure and force the network’s self-healing process. Node D was initially active and registered as a neighbor through HELLO beacons, then disconnected to simulate an unexpected network failure. In this scenario, the only admissible route is A-B-C-E. Packets that had already selected node D as their next hop before the failure may initially attempt A–B–D–E; the B–D link becomes invalidated due to the simulated failure, triggering reconvergence to A–B–C–E. The packets successfully recovered after the self-healing process were considered for the PRR calculation, while the packets that were routed via the admissible path (C) from the start were marked Not Applicable (NA).

During tests, some repetitions recorded complete delivery, while others experienced losses or required recovery. Table 9 details the results per package, including recovery status.

From the processed results, the mean PDR was 90%, the mean retransmissions were 15 per repetition, and the mean PRR was 88.33% as presented on Table 10.

#### 3.1.3. Scenario 3: Urban Context Bidirectional Communication

The same topology and bidirectional communication as in previous scenarios were maintained, but the nodes were deployed in a physical environment with a greater distance between them. Five repetitions were performed. As in Scenario 1, the admissible routes are A–B–C–E or A–B–D–E; nodes C and D are the only interchangeable hops and are parallel (equivalent) options, so tie-breaking among valid neighbors at run time does not affect the interpretation of delivery or retransmissions.

Delivery success rates varied significantly between repetitions, with some achieving full delivery and others experiencing losses. Table 11 provides the raw per-packet records.

After processing, the mean PDR was 75% and the mean retransmissions were 12 per repetition, as shown in Table 12.

### 3.2. Comparative Results

#### 3.2.1. Packet Delivery Ratio (PDR)

To assess the communication performance of the system under different conditions, the PDR was measured and compared in the three defined scenarios, taking into account the results of each of the five test runs. This approach allowed for the identification of potential performance variations due to changes in topology or environmental factors.

As illustrated in Figure 8, Scenario 1 maintained a high level of consistency, achieving 100% PDR in four out of five repetitions, with only a single repetition showing a reduced value of 75%. Scenario 2 showed a similar trend, with three repetitions reaching 100% and the remaining two presenting 75% delivery. In contrast, Scenario 3 exhibited greater variability, with 100% PDR in only two repetitions, while two others dropped to 50%, and one remained at 75%.

In addition, the red dashed line indicates a 50% threshold, included as a visual reference to highlight that none of the scenarios dropped below this value in any repetition.

These results show that a better delivery performance was observed in Scenarios 1 and 2, where conditions were ideal or included a controlled failure and recovery mechanism. In contrast, Scenario 3, which simulates a dynamic urban deployment, showed a considerable reduction in PDR between repetitions.

#### 3.2.2. Total Retransmissions

The total number of retransmissions was recorded in the three defined scenarios over five repetitions to observe how often the system required additional transmission attempts to deliver packets.

As shown in Figure 9, Scenario 1, which corresponds to ideal conditions and absence of node failure, presented the lowest overall retransmission values, ranging from 6 to 10 retransmissions over the five repetitions. Scenario 3, which shares the same network topology as Scenario 1 without node failures but with larger distances between nodes, exhibited a moderately higher range of retransmissions, fluctuating between 8 and 16.

Scenario 2 consistently presented the highest number of retransmissions among all scenarios. It reached a maximum of 18 retransmissions in the second repetition and a minimum of 10 in the fifth. In particular, this scenario includes a node failure event as part of its test conditions.

#### 3.2.3. Packet Recovery Ratio (PRR)

The PRR was evaluated exclusively in Scenario 2, as the packet recovery mechanisms are only activated under conditions of node failure or topological change.

As shown in Figure 10, in three of the five repetitions, the system achieved a complete recovery with a PRR of 100%. One repetition reached a recovery value of 75%, while the lowest observed PRR was 66.67%. The red dashed line indicates a threshold set at 65%, included as a visual reference to highlight that the system never fell below this limit during any test. Recovery ratios less than this value were not recorded.

These values reflect the data related to the recovery behavior observed under controlled failure conditions in Scenario 2.

## 4. Discussion

The evaluation of the system in three different scenarios enabled assessing its performance under controlled conditions, urban environments, and environments that involve physical network failures. Through the analysis of the PDR, PRR, and total retransmissions, the levels of reliability, effectiveness, adaptability, and resilience were determined.

### 4.1. Packet Delivery Ratio (PDR) Analysis

The PDR, defined as the proportion of successfully received packets relative to those sent, is a fundamental metric to evaluate the effectiveness and reliability of a communication system [41,42]. As shown in Figure 8, Scenario 1 achieved the highest average PDR of 95%, indicating highly reliable communication. This can be attributed to favorable testing conditions: all nodes were placed close to each other, no physical obstacles were present between them, and no known sources of interference were nearby.

In Scenario 2, where a node failure was deliberately introduced, the average PDR dropped slightly to 90%. This modest decrease reflects that while node failures do impact communication, the effects can be mitigated when environmental conditions such as short distances and the lack of obstructions remain ideal. The mesh routing protocol maintained delivery with minimal disruption.

In contrast, Scenario 3 recorded the lowest average PDR, at 75%. This scenario involved greater distances between nodes and occurred in a less controlled urban environment, where buildings, walls, and other materials likely affected transmission. Furthermore, the presence of potential interference sources (e.g., Wi-Fi networks, electronic devices, cellular antennas, or reflective surfaces) could not be ruled out. As also observed by Choi, Lee, and Lee [43], longer transmission distances tend to reduce the PDR, confirming the behavior observed in this study. However, it is important to note that in the five repetitions, the PDR never dropped below the 50% threshold, demonstrating robustness in the five runs even under challenging conditions.

These results emphasize the sensitivity of LoRa communication to environmental factors. As observed, optimal packet delivery is achieved when the nodes are near and not obstructed, while increased distance, physical barriers, and radio interference tend to degrade performance. As established by Theissen, Kianfar, and Clausen [44], radio wave propagation is influenced by multiple phenomena depending on surrounding materials, including reflection, scattering, diffraction, refraction, absorption, and shadowing. These interactions may collectively contribute to signal degradation in urban or underground deployments.

However, a detailed examination of the results per packet reveals that protocol-level factors also contribute significantly to packet losses, even under ideal conditions: based on the observation notes detailed in Section 3 tables, the observed packet losses in Scenarios 1 and 2, despite optimal short-distance conditions, result from composite causes affecting both ACK and HELLO packet delivery. Packet drops occur when ACK attempts suffer collisions or arrive late, causing ACK timeout expiration and subsequent neighbor purging by the sender. Additionally, next-hop neighbors can be purged due to missed HELLO updates caused by HELLO packet collisions or delays, as HELLO packets implement no retry mechanism.

This represents normal protocol behavior under topological constraints rather than protocol malfunction. Nodes A, C, and D become susceptible to packet drops when their next-hop neighbors become unreachable or are purged, as the loop-prevention mechanism prohibits backward forwarding to the original sender. With limited alternative routes available, packets are discarded when no valid next-hop remains.

This packet discarding occurs specifically because these nodes lack alternative routing options. Neighbor nodes can be discarded from the routing table either when ACK retries are exhausted due to collisions or delays or when neighbor tables are purged because HELLO packets never arrive or are delayed due to collisions. This occurs due to the lack of alternative routes and can generate situations where several packets may have the same number of retries but achieve different outcomes: one successfully receives an ACK and is delivered, while another with identical retry attempts is discarded because it never received a valid ACK within the timeout window and no valid neighbors are available.

Since simultaneous transmissions are less probable under ideal conditions and short distances, bidirectional traffic was deliberately implemented to test the system, recognizing that multiple packet types would coexist beyond nominal DATA flows. The actual network load significantly exceeds the nominal 2 packets per direction due to back-to-back bidirectional transmissions creating overlapping traffic. Each DATA packet triggers hop-by-hop ACKs, periodic HELLO beacons, and potential retransmissions, while these diverse packet types keep nodes in busy processing states. In general, the presence of heterogeneous traffic (different packet types) tends to depress overall performance. This aligns with previous LoRaWAN evidence showing that when confirmed and unconfirmed traffic coexist, ACK transmissions add interference and increase the likelihood of collisions [45].

It is important to note that in dense urban settings, the coexistence of multiple Low-Power Wide-Area Network (LPWAN) technologies within the same sub-GHz spectrum can affect network performance [16]. The operation of other LoRaWAN networks, the use of intermediate nodes for multi-hop communication, and devices from different operators can generate additional interference, creating challenges to prevent signal degradation and avoid collisions. This situation becomes even more complex when other sub-GHz technologies, such as SigFox, IEEE 802.15.4g, and IEEE 802.11ah, share the same frequency bands, leading to greater interference [16].

To mitigate such interference-driven PDR degradation in future deployments, a practical option is to diversify the frequency footprint of consecutive transmissions. Channel hopping, sending consecutive packets on different frequencies, helps combat losses by improving reliability, mitigating persistent interference, and reducing burstiness for more predictable delivery times. Because each channel experiences different fading and interference conditions, rotating between channels decorrelates errors and reduces the chance that successive packets face the same impairment [46].

Furthermore, the 75% average PDR observed in Scenario 3 is closely aligned with the findings of Rademacher et al. [47], who reported a similar packet reception rate of 72% in a mobile urban LoRa network deployment. Despite mobility and interference, their results were considered acceptable. Therefore, the performance observed in this study, particularly under real-world constraints, validates the resilience and effectiveness of the implemented communication system, even in the presence of node failures and dynamic network conditions.

### 4.2. Total Retransmissions Analysis

The total number of retransmissions serves as a complementary metric to assess the behavior of the system in conditions where packets are lost, delayed, or affected by adverse factors. Rather than indicating communication failure, retransmissions reflect the effectiveness of the delivery assurance mechanism, which enhances reliability by allowing retries in response to packet collisions, obstacles, signal degradation, or interference.

In Scenario 1, the system recorded the lowest number of retransmissions, with an average of only 8 per test. This outcome reflects the optimal performance of the communication link under stable conditions with short distances between nodes, without physical obstacles, and no node failures. Any retransmissions observed in this scenario probably resulted from minor delays or occasional collisions, which are typical in wireless communication.

In Scenario 2, the average number of retransmissions nearly doubled, reaching 15 per test. This increase corresponds to the intentional introduction of a node failure. The system responded appropriately by attempting multiple retransmissions before ultimately removing the unreachable node from the routing table and initiating the self-healing process. Importantly, these retransmissions were limited to the failed link, avoiding unnecessary traffic through the rest of the network. This behavior illustrates the robustness and containment strategy of the protocol under node failure conditions.

Scenario 3 also showed a relatively high average of 12 retransmissions. Although this is lower than Scenario 2, it is still about 50% higher than the ideal conditions of Scenario 1. The elevated number of retries can be attributed to external factors present in the urban deployment environment. According to Guo et al. [48], noise interference in urban settings can introduce bit errors that lead to packet erasure, prompting the system to perform additional retransmissions.

Moreover, the examination of the observation notes in the Section 3 tables reinforce that retransmissions represent a design feature rather than protocol or parameter failure. The primary underlying cause for retransmissions is ACK Timeout, occurring when ACK packets either suffer collisions or arrive delayed beyond the timeout threshold. The protocol adopts a pessimistic approach: even if an ACK is merely delayed but arrives after timeout expiration, it is considered lost and triggers retransmission. This conservative behavior prioritizes delivery assurance over efficiency.

This timeout-based approach aligns with established wireless networking principles, such as those implemented in LoRaWAN Class A devices, where acknowledgments must be received within predefined reception windows or packet loss is assumed [49]. In addition, the self-healing mechanism particularly depends on this retransmission strategy, as it must first exhaust all retry attempts on the current path before switching to an alternative next-hop neighbor, explaining the higher retransmission counts observed during failure recovery scenarios.

Furthermore, as described by Theissen, Kianfar, and Clausen [44], environmental materials and structures can affect signal propagation through scattering, absorption, and reflection, further increasing the likelihood of retransmissions. In addition, retransmissions themselves can increase the probability of packet collisions, as each attempt is effectively treated as a new transmission within the network [50].

Beyond retry-driven collisions, LBT policies can introduce synchronization pathologies that increase retransmissions and delay. LBT entails synchronization risks because nearby ongoing transmissions may go undetected (hidden terminals), while energy sensed from non-interfering directions can overprotect by causing unnecessary delays [51]. These effects can be mitigated with adaptive backoff that expands as busy-channel detections or ACK-failure rates increase and contracts otherwise, following Xanthopoulos et al. [52]. The same policy can also adapt the LBT listen duration.

As also observed by Choi, Lee, and Lee [43], weather conditions and environmental factors such as high humidity or the presence of vegetation can reduce the delivery rate of packets. These conditions can indirectly lead to a higher number of retransmissions, suggesting that future studies could analyze climatic variables to better understand their influence on communication performance.

Depending on the specific conditions of the network, such as obstacles, node failures, long distances, or signal alterations, packet losses or collisions can occur more frequently, leading to an increase in retransmissions. Moreover, this metric will likely increase in denser networks, as highlighted by Cotrim and Kleinschmidt [16], who note that when the traffic load or node density increases, network performance can be severely affected.

To further reduce losses and long ACK latencies on longer or variable-quality links, a complementary control is adaptive SF, which in turn lowers retransmissions. In particular, Adaptive Spreading Factor Selection (ASFS) enables a single-channel LoRa modem to adjust the SF by detecting the packet preamble and synchronizing the receiver to the transmitter SF without extra handshakes, which can reduce delays and improve link performance [53]. Although not implemented in this prototype, ASFS is a practical control for future deployments and remains compatible with the lightweight firmware constraints of the system.

In general, the number of retransmissions does not inherently indicate a failure. Rather, it reflects the reliability mechanism of the system, as retransmissions are intended to ensure successful delivery by awaiting acknowledgments and resending lost packets [54]. However, this metric can serve as an early indicator of abnormal network behavior, particularly if analyzed with a declining PDR that suggests network degradation or external disruption [55].

It should be noted that this study was conducted under low node density and light traffic conditions. Therefore, to better determine the relationship of this metric with network health or to enable early detection of communication issues, further tests should be performed under higher traffic loads and with a greater number of nodes.

### 4.3. Packet Recovery Ratio (PRR) Analysis

The PRR was evaluated exclusively in Scenario 2, as it was the only case in which packets initially routed to a failed node were recovered and redirected through the self-healing mechanism. PRR quantifies the proportion of these packets that were successfully rerouted and delivered after the failure occurred.

In this scenario, the average PRR was 88.33%, indicating that the system was able to recover and reroute the majority of affected packets. In particular, three out of five test repetitions demonstrated complete recovery, highlighting the effectiveness of the self-healing mechanism when the network topology changes or a node becomes unavailable. As shown in Figure 10, the PRR never fell below the 66.67% threshold in any test run, reinforcing the robustness of the system under fault conditions.

It is important to clarify that for this calculation, only packets originally routed to the failed node were considered. Due to the system’s randomized routing behavior implemented to avoid oversaturating individual nodes, some packets were fortuitously directed to valid alternative paths from the outset and therefore excluded from the PRR metric.

These results demonstrate the resilience of the system to infrastructure-level failures. The ability to dynamically redirect traffic in response to node failure is a critical feature in mesh networks, especially those deployed in environments where reliability is essential.

This aligns with the theoretical aspects discussed by Rullo, Serra, and Lobo [56], who describe self-healing mechanisms as comprising two main phases, detection and reconfiguration. In the presented system, the detection phase occurs when a neighbor node stops responding and is marked as invalid, leading to its removal from the routing table, effectively identifying the failure. The reconfiguration phase then follows, involving the rerouting of the packet that could not be delivered by the failed node through an alternative valid link.

However, the designed recovery mechanism has limitations. Its success depends on the existence of alternative routes or a sufficiently high node density. Furthermore, due to the design of the neighbor-based routing algorithm, which selects the next hop based on a randomized choice among candidates with the best RSSI-age score, the nodes do not store complete end-to-end routes. This design trade-off introduces the possibility that rerouted packets may follow longer paths, exhaust their Time-to-Live (TTL), or be discarded entirely.

This architectural decision to use a lightweight, neighbor-based approach without full routing tables was made to accommodate the hardware limitations of commercial microcontrollers. More complex strategies, such as on-demand routing or proactive route broadcasting, were avoided due to the excessive control overhead they impose, a trade-off also discussed in the work of Elliott and Heile [57].

In general, the presented approach achieves a practical balance between simplicity and effectiveness. This is supported by the high average PRR of 88.33%, which confirms the system’s ability to maintain reliable communication despite node-level disruptions.

It is important to note that while the ability to perform self-healing contributes to improving network reliability, it does not eliminate the inherent challenges associated with the reliability and availability of IoT devices such as microcontrollers. Wireless communications, typical of IoT deployments, can be less reliable than wired counterparts, and networks remain susceptible to environmental variability and even cybersecurity threats [56].

## 5. Limitations and Threats to Validity

### 5.1. Controlled Neighbor Visibility (Whitelist)

During the experiments, dynamic neighbor discovery (HELLO, expiration, and metric refresh), the routing and next-hop selection logic remained active. To obtain repeatable multi-hop conditions without unintended asymmetric links, a neighbor-visibility mask (compile-time whitelist) was applied that filters routing eligibility after discovery. Multiple peers may still be discovered via HELLO, but only whitelisted peers are eligible as next-hop candidates. This stabilizes the logical topology while keeping discovery, attachment, expiry, metrics, and routing decisions dynamic.

It is acknowledged that constraining eligibility reduces the superset of admissible paths and may attenuate short-term variability in neighbor availability. To preserve diversity under controlled conditions, the whitelist was deliberately configured to retain at least two parallel multi-hop alternative routes within a designated sub-region, enabling controlled failover (temporary node disablement) and observation of retransmissions and recovery. Similar control strategies where topology is constrained through configuration to ensure reproducibility in mesh evaluations have been reported in the literature [58].

Furthermore, the firmware exposes a configuration switch in config.h that allows the visibility mask to be disabled, enabling full dynamic discovery without further code changes. In future work, a replication study is described in full open discovery (with the same placement and metrics), along with a sensitivity analysis over HELLO and neighbor expiration intervals, to quantify the effects on PDR/PRR and retransmissions.

### 5.2. Node Density and Manual Traffic Injection

The evaluation used a five-node partial mesh with bidirectional transmissions and five independent runs per scenario. Packets were manually activated on the serial console at low rates to favor control and repeatability. For each run, bidirectional exchanges were triggered as near-simultaneous pairs two packets per endpoint, issued back-to-back within the same time window (no deliberate pauses) to create opposite-direction traffic under controlled conditions. This setup does not emulate bursty application workloads, concurrent sources, or background co-traffic. Node density was constrained by equipment procurement and availability in the deployment context, as well as budget and time, limiting the number of deployable endpoints within the study window.

With only five nodes, topology and link diversity are limited, path-length distributions are constrained, channel contention is lower, simultaneous transmissions are rarer, and LBT is exercised less often. To partially counter this, the near-simultaneous bidirectional pairs were used to force overlapping activity within the same time window. In addition to DATA packets, each hop performs an ACK per DATA packet, periodic HELLO messages support discovery and maintenance, and retransmissions may occur, all contributing to channel occupancy. Even so, these conditions can inflate PDR and deflate retransmissions relative to denser deployments, providing fewer opportunities to stress routing changes and duplicate suppression.

As future work, this study will be extended by increasing node count and spacing to 50 to 100 nodes, introducing automated scripted multi-source traffic to create overlapping, bidirectional flows at controlled rates, and adding more repetitions per scenario. Keeping placement and radio settings unchanged will allow direct comparisons with the current baseline and enable quantification of impacts on PDR/PRR, retransmissions, LBT/backoffs, and path diversity.

### 5.3. Single Physical-Layer Setting and Lack of Energy Measurements

All experiments used a single radio configuration across nodes (SF7; BW 125 kHz; CR 4/5), although the system supports other configurable radio settings via config.h, and the link–budget summary corresponds to that baseline. Energy consumption was not measured because it was outside the scope of this work. The experimental objective was to validate the minimal stack, end-to-end communication, and self-healing under controlled conditions, so no energy measurements were recorded. Therefore, it cannot be determined whether the system is energy efficient based on the present data.

Using a single physical-layer setting limits generalization across spreading factors, bandwidths, and coding rates. Different settings change range, data rate, and link robustness, which in turn can affect observed PDR, retransmissions, and channel occupancy. Because energy was not measured, delivery performance cannot be related to consumption, and range–energy trade-offs cannot be quantified. Likewise, the achievable range with alternative settings remains uncharacterized in this evaluation.

To address this, future work will replicate the same scenarios while systematically varying the physical-layer parameters defined in config.h and will instrument power to capture TX, RX, and idle states synchronized with per-packet logs. Reporting energy per delivered packet and per useful bit, along with range and retransmissions, will enable quantification of range–energy trade-offs and the identification of configurations that are more energy-efficient or that extend range without compromising reliability, with topology, traffic, and firmware otherwise held constant for fair comparison. This analysis will be particularly valuable for subsequent iterations targeting real-world deployments at scale or battery-powered mobile nodes, where energy consumption is a first-order design constraint.

## 6. Conclusions

This work demonstrates that LoRa mesh networking without gateways is feasible on commodity ESP32 microcontrollers, providing a foundation for decentralized IoT communication research. The prototype implementation validates the key contributions presented: a gateway-free mesh implementation operating directly on ESP32 hardware; a lightweight protocol featuring neighbor-based routing and hop-by-hop acknowledgments; self-healing mechanisms that automatically recover from node failures; and empirical validation across test scenarios. While this prototype-scale five-node demonstration lacks production features such as security and energy optimization, it establishes the viability of the core design and provides an open-source foundation for further development.

Three scenarios were evaluated, namely ideal conditions, sudden node failure, and urban deployment. Under ideal conditions, the system achieved a PDR of 95% with minimal retransmissions. In node failure, self-healing recovered traffic that had initially traversed the failed node, with a PRR of 88.33% and PDR of 90%, indicating resilience. In the urban case the PDR reached 75%, which highlights susceptibility to obstacles, interference, and longer links. Retransmissions served as an indirect indicator of network health, remaining minimal under ideal conditions, nearly double after the failure event, and moderately higher in the urban scenario.

Regarding limitations and threats to validity, the experiments used firmware-locked neighbor lists that prevented free routing and reduced route diversity. The prototype involved only five nodes with manually triggered low-rate traffic. All tests were performed under a single radio configuration (SF7; 125 kHz; CR 4/5). Energy consumption was not measured, so energy efficiency cannot be claimed.

Future work will focus on several concrete steps. One is to conduct scale-out experiments with 50 to 100 nodes, add energy profiling for transmit, receive, idle, and sleep modes, and evaluate adaptive spreading factor policies to quantify the range, energy, and reliability trade-offs. The other is to perform a baseline comparison against Meshtastic and LoRaBlink under identical radio settings and topologies using common metrics such as PDR, PRR, latency, retransmissions, and energy. Finally, a controlled, like-for-like baseline comparison will be conducted against representative LoRa mesh implementations under identical radio settings and topologies, reporting PDR, PRR, latency, retransmissions, and energy per delivered bit. This campaign will be run on the same hardware with shared instrumentation to ensure reproducibility.

## Figures and Tables

**Figure 1 sensors-25-06036-f001:**
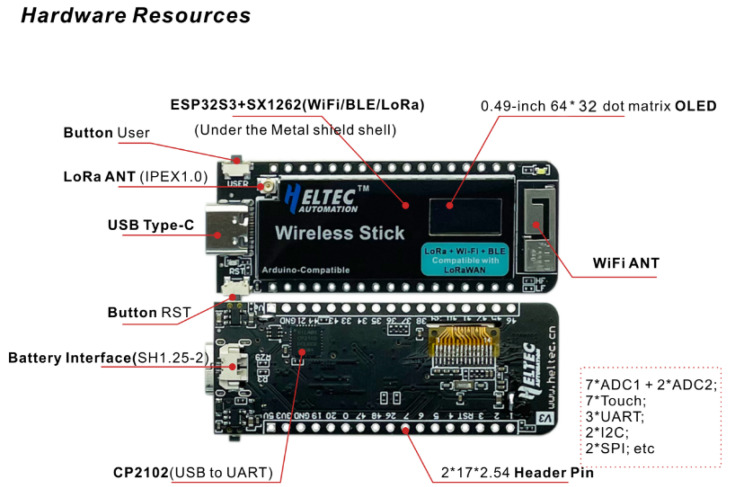
PinOut scheme for the Wireless Stick development board. Source: Taken from the official board documentation [31].

**Figure 2 sensors-25-06036-f002:**
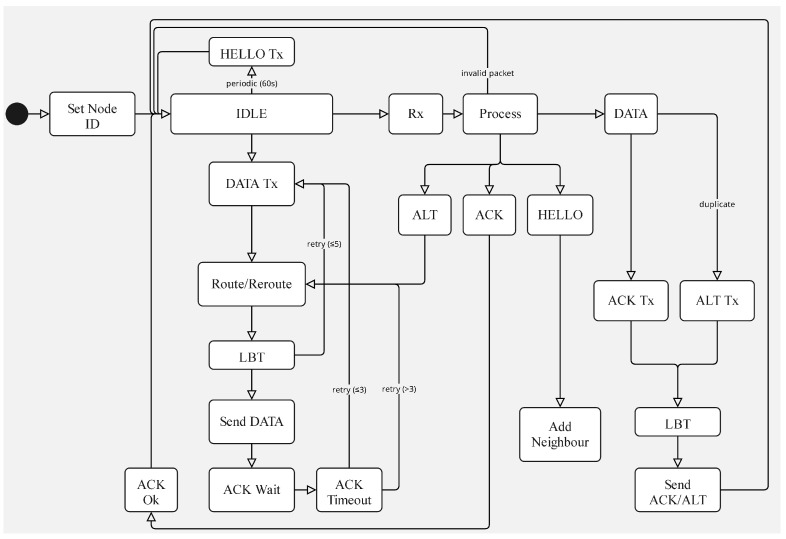
State machine diagram illustrating the general operation of the mechanisms and their interrelation within the system.

**Figure 3 sensors-25-06036-f003:**
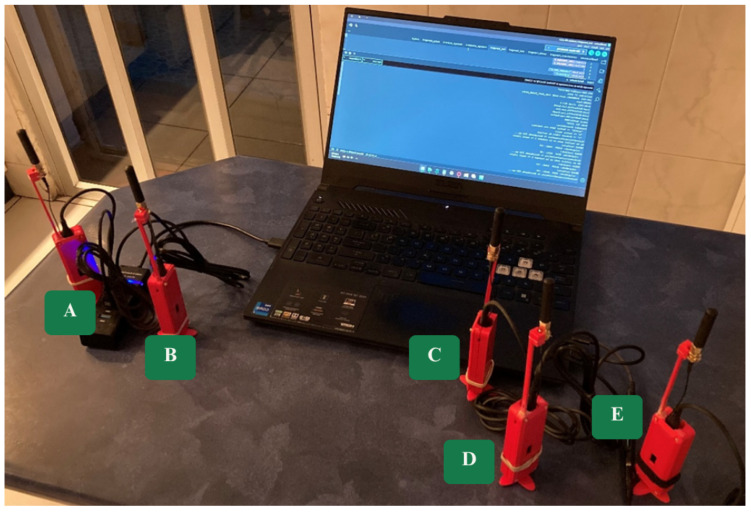
Final implementation of the 5-node mesh prototype. Each node was physically assembled and loaded with the custom firmware.

**Figure 4 sensors-25-06036-f004:**
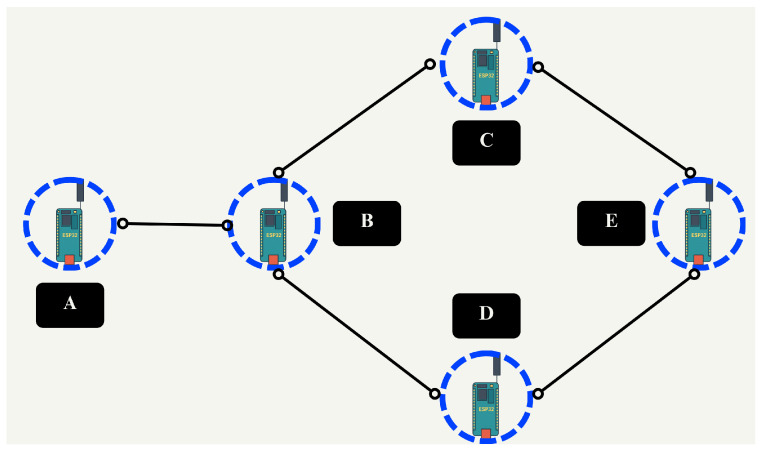
Implemented network topology. The design includes multiple-hop communication and alternative routing paths.

**Figure 5 sensors-25-06036-f005:**
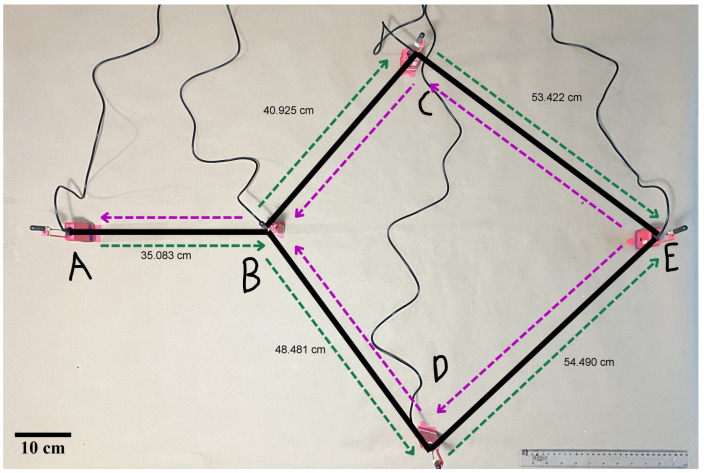
Ideal conditions scenario. Nodes are deployed close to each other, enabling multi-hop and bidirectional communication with redundant paths.

**Figure 6 sensors-25-06036-f006:**
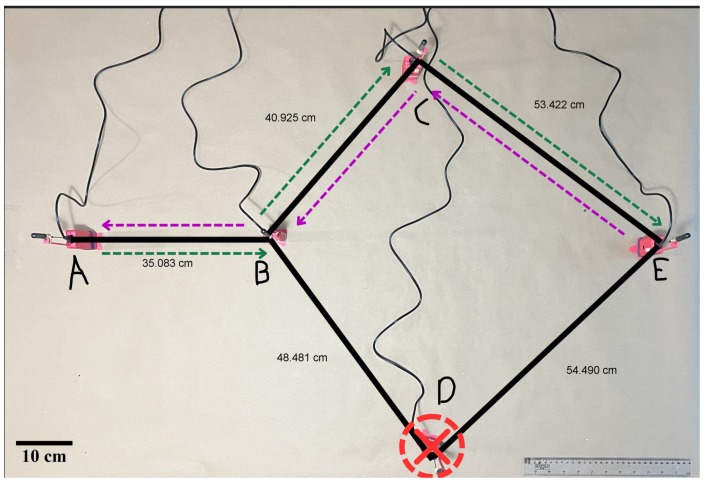
Node-failure scenario. Node D is disconnected, but communication between A and E remains functional through alternative routes.

**Figure 7 sensors-25-06036-f007:**
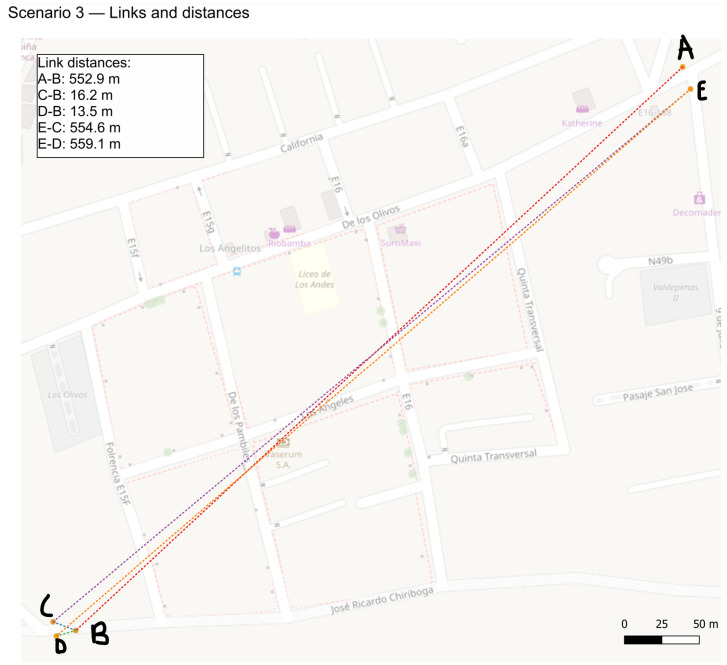
Physical deployment of the nodes and estimated inter-node distances, visualized using QGIS.

**Figure 8 sensors-25-06036-f008:**
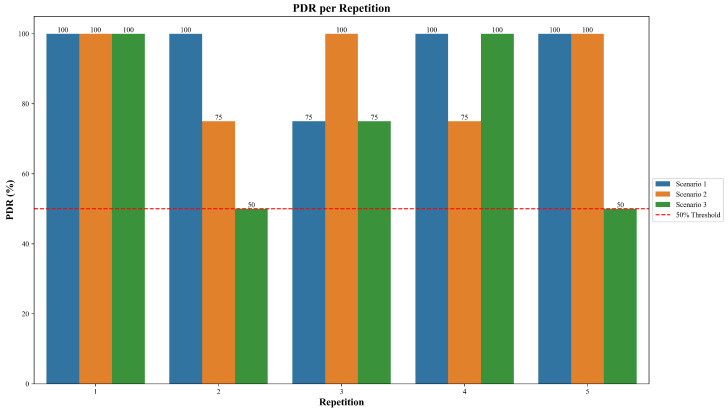
Comparative Packet Delivery Ratio (PDR) analysis across the three testing scenarios.

**Figure 9 sensors-25-06036-f009:**
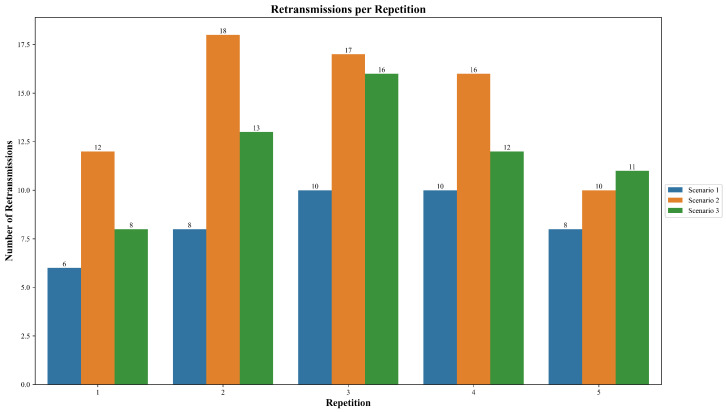
Comparative analysis of total retransmissions across the three testing scenarios.

**Figure 10 sensors-25-06036-f010:**
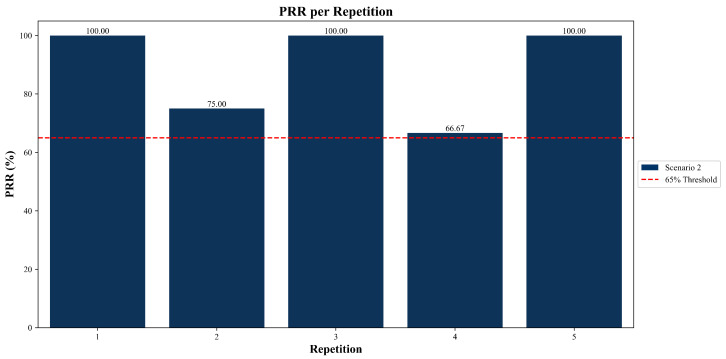
Packet Recovery Ratio (PRR) results across all five test repetitions in Scenario 2.

**Table 1 sensors-25-06036-t001:** Comparison between the present system and representative prior works.

Work/ System	Target Hardware	Routing	Reliability	Channel Access	Code	Eval. Type	Energy Reporting	Ref.
INET LoRa	SX127x	RSSI-based optimal path	No	ALOHA	Not available	Simulation	No	[29]
LoRaBlink	STM32 + SX1272	Flooding + Slotted Tx	ACK	Slotted ALOHA	Open-Source	Field	Yes	[27]
Meshtastic	ESP32/nRF52840/RP2040 + SX1262/SX127x/LR11xx	Flooding	ACK	LBT	Open-Source	Field	No	[28]
This work	ESP32-S3 + SX1262	Neighbor-based	ACK	LBT	Open-Source	Field	No	–

**Table 2 sensors-25-06036-t002:** Specifications of the Heltec Wireless Stick V3 (ESP32-S3 + SX1262) development board, adapted from Heltec official documentation [31].

Parameter	Description
Master Chip	ESP32-S3FN8 (Xtensa^®^ 32-bit LX7 dual-core, up to 240 MHz)
LoRa Transceiver	SX1262, high sensitivity (−134 dBm)
Frequency Bands	470–510 MHz and 863–928 MHz
Max. TX Power	21 ± 1 dBm
Display	0.49-inch OLED (64 × 32 dot matrix)
Memory	384 KB ROM, 512 KB SRAM, 16 KB RTC SRAM, 8 MB SiP Flash
Connectivity	Wi-Fi 802.11 b/g/n (up to 150 Mbps), Bluetooth 5 and Bluetooth Mesh
Interfaces	Type-C USB; 2 × 1.25 battery connector; IPEX LoRa antenna port; 2 × 17 GPIO header
Peripheral Support	7 × ADC1, 2 × ADC2, 7 × Touch, 3 × UART, 2 × I^2^C, 2 × SPI
Power Management	SH1.25 battery interface, integrated lithium battery management (charging, protection, auto-switching)
USB Interface	CP2102 USB to serial chip with ESD and short-circuit protection
Operating Temperature	−20 to 70 °C
Dimensions	58.08 × 22.6 × 8.2 mm

**Table 3 sensors-25-06036-t003:** Glue-rod antenna technical summary adapted from Heltec official documentation [32].

Parameter	Description
Type	Glue-rod monopole
Standing Wave Ratio (SWR)	≤2.0
Antenna gain	3 dBi
Polarization	Linear
Impedance	50 Ω
Radiator material	Cu (copper)
Plastic body	PC + ABS
Connector pull test	≥2 kg
Operating temperature	−40 °C to +65 °C
Storage temperature	−40 °C to +80 °C
Connector/cable	SMA male via IPEX (U.FL) Ver.1 to SMA feeder (100 mm)

**Table 4 sensors-25-06036-t004:** Description of firmware source files and their functionality.

File	Functionality
LoRaMesh.ino	Initializes the system and orchestrates the main execution flow.
lora_manager.h	Configures the SX1262 LoRa transceiver with frequency, power, and callbacks.
oled_manager.h	Controls the OLED screen for displaying status and debug messages.
communication_manager.h	Handles the transmission and reception of packets.
routing_manager.h	Determines the next hop based on the neighbor table.
packet_manager.h	Defines and parses DATA, ACK, HELLO, and ALT packets.
message_scheduler.h	Queues outgoing messages and manages retries and LBT timing.
message_receiver.h	Processes received packets and handles duplicates and ACKs.
config.h	Central configuration of frequency, power, mesh ID, retry limits, etc.

**Table 5 sensors-25-06036-t005:** Radio configuration parameters used for all nodes in the prototype.

Parameter	Value
LoRa Frequency	915 MHz
Bandwidth (BW)	125 kHz
Spreading Factor (SF)	7
Coding Rate (CR)	4/5
Transmission Power	22 dBm
Preamble Length	8 symbols
Cyclic Redundancy Check (CRC) Enabled	Yes
In-Phase/Quadrature (I/Q) inversion	Standard (not inverted)
Listen Before Talk (LBT)	Enabled
Retry Limit per Hop	3 attempts

**Table 6 sensors-25-06036-t006:** Single-link budget at 915 MHz (SF7 baseline). Formulas in Appendix A.

Parameter	Value
Frequency	915 MHz
Bandwidth (BW)	125 kHz
Spreading Factor (SF)	7
Coding Rate	4/5
Transmit power	22 dBm
TX feeder/connector loss	~0.33 dB
TX antenna gain	+3 dBi
Effective Isotropic Radiated Power (EIRP)	24.67 dBm
RX antenna gain	+3 dBi
RX feeder loss	~0.33 dB
Receiver sensitivity (SF7, 125 kHz, CR 4/5)	−124 dBm
Budgeted Implementation loss	2 dB
Target fade margin	20 dB
Nominal Maximum allowable path loss (MAPL)	~151.35 dB
Usable Maximum Allowable Path Loss (MAPL)	~129.35 dB

**Table 7 sensors-25-06036-t007:** Transmission records in Scenario 1 (per-packet).

Run	Direction	Sent	Received	Retrans.	Note
1	A→E	X	X	0	
		X	X	3	AT
	E→A	X	X	2	AT
		X	X	1	AT
2	A→E	X	X	3	AT
		X	X	0	
	E→A	X	X	4	AT
		X	X	1	AT
3	A→E	X	X	4	AT
		X	—	3	AT + D
	E→A	X	X	0	
		X	X	3	AT
4	A→E	X	X	0	
		X	X	2	AT
	E→A	X	X	5	AT
		X	X	3	AT
5	A→E	X	X	1	AT
		X	X	3	AT
	E→A	X	X	0	
		X	X	4	AT

X = packet received; — = packet lost; AT = Ack Timeout; D = Dropped Packet.

**Table 8 sensors-25-06036-t008:** Average PDR and retransmissions in Scenario 1.

Run	PDR (%)	Retrans.
1	100.00	6
2	100.00	8
3	75.00	10
4	100.00	10
5	100.00	8
Mean	95.00	8.00

Values averaged over the two traffic directions in each run.

**Table 9 sensors-25-06036-t009:** Transmission records in Scenario 2 (per-packet).

Run	Direction	Sent	Received	Recovered	Retrans.	Note
1	A→E	X	X	NA	1	AT
		X	X	NA	0	
	E→A	X	X	X	4	SH + AT
		X	X	X	7	SH + AT
2	A→E	X	X	X	4	SH + AT
		X	X	X	7	SH + AT
	E→A	X	X	X	4	SH + AT
		X	—	—	3	AT + D
3	A→E	X	X	X	4	SH + AT
		X	X	X	6	SH + AT
	E→A	X	X	NA	0	
		X	X	X	7	SH + AT
4	A→E	X	—	—	0	AT + D
		X	X	X	7	SH + AT
	E→A	X	X	NA	1	AT
		X	X	X	8	SH + AT
5	A→E	X	X	NA	0	
		X	X	NA	2	AT
	E→A	X	X	X	4	SH + AT
		X	X	X	4	SH + AT

X = packet received; — = packet lost; AT = Ack Timeout; D = Dropped Packet; SH = Self-Heal; NA = not applicable (packet was already routed through the alternate path).

**Table 10 sensors-25-06036-t010:** Average PDR, retransmissions, and PRR in Scenario 2.

Run	PDR (%)	Retrans.	PRR (%)
1	100.00	12	100.00
2	75.00	18	75.00
3	100.00	17	100.00
4	75.00	16	66.67
5	100.00	10	100.00
Mean	90.00	15.00	88.33

PRR calculated only for packets recovered after the failure event.

**Table 11 sensors-25-06036-t011:** Transmission records in Scenario 3 (per-packet).

Run	Direction	Sent	Received	Retrans.	Note
1	A→E	X	X	3	AT
		X	X	2	AT
	E→A	X	X	2	AT
		X	X	1	AT
2	A→E	X	X	3	AT
		X	—	0	AT + D
	E→A	X	—	3	AT + D
		X	X	7	AT
3	A→E	X	X	3	AT
		X	X	8	AT
	E→A	X	X	2	AT
		X	—	3	AT + D
4	A→E	X	X	4	AT
		X	X	2	AT
	E→A	X	X	2	AT
		X	X	4	AT
5	A→E	X	—	6	AT + D
		X	X	2	AT
	E→A	X	—	3	AT + D
		X	X	0	

X = packet received; — = packet lost; AT = Ack Timeout; D = Dropped Packet.

**Table 12 sensors-25-06036-t012:** Average PDR and retransmissions in Scenario 3.

Run	PDR (%)	Retrans.
1	100.00	8
2	50.00	13
3	75.00	16
4	100.00	12
5	50.00	11
Mean	75.0	12.00

Values averaged over the two traffic directions in each run.

## Data Availability

Dataset available on request from the authors.

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
