# Peer review of "Gateway-Free LoRa Mesh on ESP32: Design, Self-Healing Mechanisms, and Empirical Performance"

_sensors, 2025, doi:10.3390/s25196036_

Round 1
Reviewer 1 Report
Comments and Suggestions for Authors
The shortcomings of the paper are as follows:
- The main contributions of the paper are unclear, and the authors need to list them one by one. 2. The designed scheme is not detailed and specific enough.
- Some formulas are not standard enough.
- Some key formulas in the paper have been ignored.
- The research content of the paper is too complicated, and the authors should highlight the key technologies that have made substantial contributions in detail.
- During simulation, authors should compare and analyze with existing methods.
Author Response
We thank the reviewer for their comments, which have been addressed in the attached document.

Reviewer 2 Report
Comments and Suggestions for Authors
Dear Editor,
The authors of this paper present a design and evaluates the implementation of a gateway-free LoRa mesh network on affordable ESP32-S3 hardware. The approach deployed on the paper moved beyond the centralized LoRaWAN paradigm and providing a practical, open-source alternative with custom self-healing mechanisms. The work is very relevant as it address the need for decentralised, low-cost, and resilient IoT communication in infrastructure-constrained settings. The work also provides some empirical evaluations, where the design was tested under ideal, failure, and realistic urban conditions. Some key performance metrics such as Packet Delivery Ratio and Packet Recovery Ratio were usd in the analysis. The results obtained demonstrated that the proposed design can be used in ideal condition after scoring 95% PDR and 88.33% PRR in failure scenarios. Even though the work seems to be novel and interesting worth accepting for publications, it is worthy noting some limitations that could be addressed by the authors, thus:
- This is the most critical omission in the proposed design is lack of security. It was explicitly states in Page 7, lines 219 to 233 that “Finally, it is important to note that this system currently does not incorporate security mechanisms. Neither encryption nor authentication is applied to the transmitted packets” The lack of encryption, authentication, or integrity checks makes the unsuitable for any real-world application where data confidentiality or node authenticity is important. This now underscores the earlier result of obtained of 95% PDR, suggesting its applicability in ideal conditions. I feel this is a major limitation the authors should note, even though it has been acknowledged.
- Another limitation of this paper is the used of Fixed Topology and fewer number of nodes (See page 23-24). The deployment of fixed topology would certainly limits the evaluation, as it doesn't test the system's dynamic discovery and routing capabilities, which is a peculiar setting for a typical ad-hoc networks. Moreover, in the experiment, only a fraction i.e. 5 units were used. This would naturally triggered, very low-rate traffic, hence, stress test of the system's performance were not assessed under heavy load. Therefore, the work could have benefitted more by conducting larger-scale experiments with perhaps 100 nodes and above automated to test the scalability of the design. Other contending issues include the use of single physical layer, lack of energy consumption measurements in the reports and finally the work was not compared with other s
Author Response

(The authors gave the same response as above.)

Reviewer 3 Report
Comments and Suggestions for Authors
The manuscript is devoted, in my opinion, to a relevant topic - the development of alternative network protocols for networks using LoRa technology at the OSI model physical level, and providing direct decentralized interaction between microcontrollers without the need for a fixed infrastructure. An additional the work advantage is the proposed protocols implementation on the currently popular ESP32 microcontroller together with the LoRa Transceiver SX1262 and experimental verification of a small network built using them in various scenarios. However, the experiment and its results description, in my opinion, contains a number of shortcomings that need to be corrected:
- Section 2.2.2 (lines 172–200) describes the mechanism for selecting neighbors and the node through which data is transmitted, stating that a faulty neighbor is removed from the neighbor list. However, Section 2.3.2 (lines 263–265) states that the neighbor list is fixed in the firmware. However, it is not explained whether the algorithm for removing faulty neighbors is implemented in the firmware and how this affects the results obtained?
- The experimental results description includes large tables (7, 9, 11) describing the each packet transmission for each of the scenarios under consideration. However, the information provided in them is not intuitively clear to me. What is the packet IDs for? Through which nodes was the information transmitted? What is the errors and retries cause? I think it would be more relevant to provide the each packet route, analyze the errors causes, and how retries were performed.
- As I understand it, only five packets were transmitted during the experiment for each scenario. I think that using statistical analysis with such a small sample size is incorrect. It seems more relevant to me to conduct a detailed packet losses analysis in the first and second scenarios. Since the distances at which transmission was carried out in these scenarios are insignificant, the reason for packet losses in these scenarios is not clear to me. I would like to be sure that the reason for these losses does not lie in the developed protocols or their parameters.
I believe that correcting these comments will significantly improve their manuscript.
Author Response

(The authors gave the same response as above.)

Round 2
Reviewer 3 Report
Comments and Suggestions for Authors
In the new manuscript version, the authors managed to correct almost all of the comments I had. There is only one request left, which I have already written about: I would like to supplement the tables (7, 9, 11) with information about the route along which each packet is transmitted, and more detailed information about the failure cause and location, if the authors have such information. For example, why in scenario 1 in the 3rd case A-E the second packet is lost after 3 retries, and in the 5th case E-A the first packet is transmitted after 5 retries. Or why in scenario 2 attempts are made to transmit data through node D, despite the appearance of the Self-Heal state.
Author Response
We welcome your comments and suggestions for improving the paper, which are addressed in the attached document.
